# Cationic UV-Curing of Epoxidized Biobased Resins

**DOI:** 10.3390/polym13010089

**Published:** 2020-12-28

**Authors:** Camilla Noè, Minna Hakkarainen, Marco Sangermano

**Affiliations:** 1Politecnico di Torino, Dipartimento di Scienza Applicata e Tecnologia, C.so Duca Degli Abruzzi 24, 10129 Torino, Italy; camilla.noe@polito.it; 2Department of Fibre and Polymer Technology, KTH Royal Institute of Technology, 100 44 Stockholm, Sweden; minna@kth.se

**Keywords:** UV-curing, epoxy resin, cationic photopolymerization

## Abstract

Epoxy resins are among the most important building blocks for fabrication of thermosets for many different applications thanks to their superior thermo-mechanical properties and chemical resistance. The recent concerns on the environmental problems and the progressive depletion of petroleum feedstocks have drawn the research interest in finding biobased alternatives. Many curing techniques can be used to obtain the final crosslinked thermoset networks. The UV-curing technology can be considered the most environmentally friendly because of the absence of volatile organic compound (VOC) emissions and mild curing conditions. This review provides an overview of the state of the art of bio-based cationic UV-curable epoxy resins. Particular focus has been given to the sources of the bio-based epoxy monomers and the applications of the obtained products.

## 1. Introduction

Epoxy resins are versatile polymer materials widely used as coatings, adhesives, matrices in composites, and electronic applications. They possess high strength, good dimensional stability, and excellent adhesion with substrates. However, the majority of them are petroleum-derived, which leads to environmental and feedstock depletion issues.

The global epoxy resin market was accounted for $7.54 billion in 2015 and was projected to be $11.22 by 2021 [1]. The most used epoxy resin is the diglycidyl ether of bisphenol-A (DGEBA), commonly cured under ambient conditions with polyamines or polyamides [2]. The DGEBA is the condensation product of bisphenol-A (BPA) and epichlorohydrin. Bisphenol-A is a reprotoxic R2 substance that was initially synthesized as a chemical estrogen. It negatively impacts wildlife and it can disrupt the human endocrine system by mimicking the human body hormones [3,4,5]. For this reason, several studies have been conducted to find alternative aromatic compounds to produce epoxy thermosets without BPA. Increasing emphasis has been addressed to the development of epoxy resins from biobased resources in line with green-chemistry principals to avoid environmental issues, and depletion of non-renewable resources.

The term “biobased” refers to a product synthesized from renewable resources and does not imply biodegradability. As a matter of fact, the increasing market interest in biobased materials and research activities are mainly focused on increasing the performances and durability instead of the biodegradability [6].

Among bio-based monomers, vegetable oils are an interesting renewable resource since they are cheap and abundant. They are composed of 90–95% triglycerides with a high number of double bonds. These unsaturated sites can be functionalized with epoxy groups. The obtained conversion is usually higher than 90% [7,8]. The aliphatic structure of the epoxidized vegetable oils (EVOs) leads to thermosets with low thermo-mechanical properties; this limits their usage alone in non-structural applications. They have been mostly used as matrices for bio-composites with cellulose [9] or as tougheners in blends with petroleum-derived epoxy resins since their flexible nature can enhance the fracture and impact properties of the thermoset materials [10,11,12]. Moreover, the obtained cross-linked networks are potentially biodegradable through the cleavage of the glycerol ester bonds [2,13].

Another renewable monomer source is the cashew nut shell liquid (CNSL), which resides in the Anacardium occidentalis mesocarp. The distillation of this liquid produces natural phenolic moieties, meta-substituted cardanols. The meta-substitution consists of C_15_ alkyl chains with different percentages of unsaturation: 8.4% of saturated chains, 48.5% mono-olefinic, 16.8% diolefinic, and 29.3% triolefinic [14]. Epoxidized cardanol derivatives are used in combination with other epoxy resin to produce new types of blends for coating applications requiring higher toughness [15,16] or for preparation of anticorrosive paints [2]. The cardanol market is currently dominated by Cardolite^®^ corporation which sells many epoxidized cardanol resins, reactive diluents and resin modifiers.

Although the biobased thermosetting materials are environmentally friendly, most of them have limitations linked to their processability such as long curing schedules and high curing temperature [17]. Moreover, most of the curing processes involve the release of volatile organic compounds (VOC).

To avoid these problems, the UV-curing technology is gaining increasing attention as a way to enhance safety and working conditions and to boost the curing processes.

There are some reviews regarding the synthesis and curing of epoxidized bio-based resins witnessing high interest towards these materials [4,6,8,18,19,20,21,22,23,24,25,26]. However, none of these reviews focused explicitly on the cationic UV-curing.

To fill the gap, the purpose of this review is to present an overview of the cationic UV-cured bio-materials and their fields of applications. In the first section, some backgrounds are recalled on the cationic UV-curing processes, including the structure of the most common photo-initiators and the description of the reactions leading to epoxy networks. The second and third sections will focus on epoxidized vegetable oils and epoxidized cardanol derivatives and their cationic UV-cured networks, respectively. The fourth section will present less common cationic UV-curable bio-based epoxidized monomers. Finally, in the conclusion section there will be some consideration on the perspectives of those materials.

## 2. Cationic UV-Curing

Crivello developed the cationic photopolymerization in 1977. Crivello discovered that diaryliodonium salts (Ar_2_I^+^MtXn^−^) could generate high acidic solutions when irradiated by UV light. The photolysis of those salts produced Brφnsted acid that could initiate the cationic chain-growth polymerization [27]. The mechanism of the cationic ring-opening polymerization of epoxides is reported in the Figure 1.

The cationic UV curing technique has many advantages over the free-radical photopolymerizations, such as:
It does not require an inert atmosphere during the curing process since it is not affected by oxygen inhibition.It can continue after the light source has been removed. This phenomenon is noted as a “dark reaction”, which can lead to enhanced monomers conversion either in ambient temperature or with a thermal treatment.The cationic photocurable monomers are generally not toxic and irritating.The obtained materials are less affected by volume shrinkage upon curing and possess higher thermal resistance [28,29].

### 2.1. Cationic Photoinitiators

The most used cationic photoinitiators (PhIs) are the onium salts. The onium salts are ionic compounds composed of an organic cation and an inorganic anion. The cationic component absorbs light, so its structure determines the photosensitivity, quantum yield, and the ultimate thermal stability of the salt. The anionic component defines the strength of the generated acid, the initiator efficiency, and the reactivity of the propagating ions pair in the polymerization reaction.

The UV irradiation produces the photoexcitation and, subsequently, the excited singlet state’s decay inducing both homolytic and heterolytic cleavages. The most used photoinitiators for photoinduced cationic ring-opening polymerization are diaryliodonium, triarylsulfonium, and dialkyphenacylsulfonium. Their structures are reported in Figure 2 where MtXn^−^ represents a weak nucleophilic counterion, such as BF_4_^−^, PF_6_^−^ or SbF_6_^−^.

The simplified photodecomposition of general diaryliodonium salts is shown in Figure 3 [30]. The cleavage of the Ar-I bond produces reactive cations, radical-cations, and radicals. Cations and radical-cations react further with proton donor species (solvents or monomers) to form Brφnsted-acids, which are the real initiating species [31]. The photolysis of those salts is an irreversible process. The acids photogenerated are called “superacids” since their acidity values in the Hammet scale (H0 values) range from −14 to −30. Higher anionic dimension leads to lower nucleophilicity, and photogeneration of stronger acids. Therefore, even if the anions have no direct role in the photochemistry, their structural dimensions strongly influence the polymerization kinetics [32,33].

Generally, the UV activation wavelength region of these photoinitiators ranges from 230 to 300 nm. In order to enhance the performances of the onium salts, different methodologies can be applied:

Shifting the activation region by introduction of chromophoric groups in the aromatic rings [34].Indirect activation of the photoinitiator by free radical oxidation [35,36], electron transfer from a photoexcited molecule and onium salts [37,38,39], or by the excitation of charge transfer complexes of the salts [40,41].

Another class of photoinitiators is constituted by dialkylphenacylsulfonium salts, that are able to withstand reversible photolysis processes. When irradiated, they form reversible yields and strong Brφnsted acids. The drawback of this reversible system is the short dark polymerization, since in the absence of UV light, the termination reaction takes place rapidly. [42,43].

Alkoxypyridinium salts on the other hand have excellent thermal stability and high solubility [44,45,46,47]. New carbazole scaffold visible light photoinitiator/photosensitizers were recently developed by Lalevée et al. These systems consist of a carbazole unit connected at two 4,4′-dimethoxydiphenylamine groups in positions 3 and 6 [48]. Other types of photosensitizers of onium salts are naphthalene-based compounds. For example, 1-Amino-4-methyl-naphthalene-2-carbonitrile derivatives have been studied by Ortyl as versatile UV and visible light sensitizers [49].

Versace et al. have studied the use of anthraquinone functional phthalocyanine in combination with suitable co-initiators as visible light photoinitiator systems for cationic and free-radical polymerization [50]. The progress in cationic photoinitiators has been in detail analyzed in recent reviews [28,29,45,46,47].

### 2.2. Cationic UV-Curable Monomers

The use of photolatent acids can be exploited to polymerize unsaturated monomers such as vinyl ethers, styrene and N-vinylcarbazole as well as the ring-opening polymerizations of epoxy monomers, cyclic ethers, lactones and cyclic acetals. These monomers can polymerize by a cationic chain growth polymerization mechanism.

Vinyl ether monomers [51,52] were first investigated as a good alternative to the chain growth radical photopolymerization of acrylates because vinyl ethers possess low toxicity and low odor as well as high reactivity. Nevertheless, under certain conditions, hydrolysis may compete with the polymerization of vinyl ether monomers. The hydrolysis reaction of vinyl ethers with water is catalyzed by the same acid species that catalyze the polymerization reactions.

The commercially most important class of cationically photocurable monomers is nowadays epoxides [53,54,55,56,57,58,59,60,61,62,63] they can undergo cation-induced ring-opening polymerization reaction through an oxiranium ion intermediate. When difunctional epoxides or epoxy-substituted polymers are used as the starting materials, crosslinking readily occurs to generate a three-dimensional polymer network.

Among the different available epoxy monomers, only cycloaliphatic epoxides have reached substantial commercial significance due to their higher reactivity in cationic photopolymerization and also due to excellent adhesion, chemical resistance and mechanical properties of the resulting thermosets.

## 3. Cationic Photocurable Bio-Based Epoxy Monomers

### 3.1. Epoxidized Vegetable Oils (EVOs)

Botanical oils are inexpensive and readily available (Figure 4). Based on the plant species they were recovered from; these oils contain different amounts of olefinic double bonds on different position. The scientific interest in these oils has continuously grown in recent years.

The synthesis of cationically UV-curable epoxy monomers from vegetable oils (VO) was first reported in 1992 by Crivello and Narayan. In this work, soybean, safflower, sunflower, canola, crambe, lesquerella, rapeseed, meadowfoam, and linseed oils were epoxidized following two different epoxidation techniques. The first epoxidation method involved the use of Amberlite IR-120 resin as a catalyst, glacial acetic acid, and hydrogen peroxide (H_2_O_2_), while the second utilized methyltrioctylammonium (diperoxotungsto)phosphate (MTP) as phase-transfer catalyst with H_2_O_2_. Vernonia galamensis oil isolated from seeds was also included because it contains natural epoxidized triglycerides. All the oils previously mentioned were photocrosslinked using different diaryliodonium and triarylsulfonium salts bearing alkoxy groups with different chains-lengths. It was demonstrated that the presence of long alkoxy-substitution increased the PhI efficiency and improved its solubility into the monomers. The epoxidized vegetable oils (EVOs) were cured under a fusion UV-light system. The fusion lamp apparatus is different from the conventional static lamp since, in this case, the specimen is placed in a moving belt, the velocity of which is controlled by the user. The curing conditions were furtherly evaluated by varying temperature, light intensity, and PhI concentrations. To obtain tack-free films under visible light irradiation, the addition of different types of photosensitizers like anthracene, pyrene, or perylene into the ELO formulations was also investigated. The thermosetting polymers obtained in this work showed good mechanical properties and adhesion to steel, aluminum, and glass [64].

More recently, another broad investigation concerning the epoxidation and cationic photocuring of 12 different vegetable oils was conducted by Malburet et al. The one step process optimized epoxidation technique involved acetic acid, H_2_O_2_, and Amberlite^®^ IR-120. This synthesis led to almost full conversion of the double bonds, the conversion ranging from 97% to 100%. The EVOs were cured using either UV or hardener-free thermal curing. Formulations containing 2 wt% of triarylsulfonium hexafuoroantimonate salts were cured under a Fusion lamp. This work demonstrated the possibility to tune the thermomechanical properties of the EVO thermosets by carefully selecting the starting vegetable oil. Higher double bond content in the raw materials resulted in higher epoxy index and improved thermo-mechanical properties [65].

In 1998, Chakrapani and Crivello described two castor oil epoxidation routes, involving different hydrogen peroxide concentrations leading to 3.6% and 3.7% oxirane contents, determined by titration. Moreover, the Castung^®^ epoxidation leading to oxiranic oxygen content of 6.7% was also reported. Castung^®^ oil is a highly unsaturated oil derived by acid treatment of castor oil. ECO has a peculiar molecular structure with respect to the other types of EVOs; since in its chain both epoxy and hydroxyl groups are simultaneously present (Figure 5). However, this structure is no longer present in the epoxidized Castung ^®^ (ECT) since the acid treatment dehydrates the chains.

The cationic UV-curing using diaryliodonium salt photoinitiators of ECO, ECT, epoxidized linseed oil, soybean oil, and synthetic epoxide resins (bisphenol A diglycidyl ether modified resin-DER 331 and 3,4-epoxycyclohexylmethyl-3′,4′-epoxy-cyclohexanecarboxylate-CY 179) were evaluated and compared. Photopolymerization kinetics were also studied either after the addition of photosensitizers or by varying the PhI structure and concentration. The influence of anthracene and 2-isopropylthioxanthone (ITX) photosensitizers showed an increase in the polymerization kinetics of ECO. Thanks to the presence of hydroxyl groups in the chain, ECO has been proven to have a higher conversion and polymerization rate than EVOs with higher epoxy content. This result can be explained considering the activated monomer mechanism (AM). The nucleophilic attack of the hydroxyl groups in the growing ionic chain-end produces a protonated ether. The termination of this growing chain by the proton transfer to a new epoxy monomer can start a new chain. The influence of hydroxyl groups in the ECO copolymerization with DER 331 or CY 179 was also investigated [66].

In 2019, the epoxidation of castor oil and its cationic photocuring using triarylsulfonium hexafluoroantimonate as a photoinitiator was investigated. The thermo-mechanical properties of the obtained films were investigated and compared with other networks deriving from different epoxidized biobased monomers. The ECO crosslinking produced a flexible polymeric network with high conversion (85%) [67]. More details on the other biobased monomers will be provided in Section 3.3.

In collaboration with Crivello, Ortiz et al., further investigated the reaction kinetics of epoxidized natural oils (epoxidized soybean, sunflower, corn, and linseed oils). In order to boost the cationic photopolymerization, they evaluated the addition of different mono, di, and trisubstituted alcohols. In this work, the epoxidation synthesis route implied the use of H_2_O_2_ and methyl trioxo rhenium (MTO) as a phase transfer catalyst. In all cases, the addition of alcohol groups increased the reaction kinetics. This result can be explained considering two different combined mechanisms called the activated monomer mechanism and the radical-induced cationic photopolymerization [68].

Recently Yang et al. studied the structure-property relationship of photocured epoxidized soybean oil and epoxidized linseed oil. They were able to costume the average epoxy functionality (fepoxy) present in the formulations either by using a chemical epoxidation of soybean oil involving H_2_O_2_ and a quaternary ammonium phosphotungstate (W-based) phase transfer catalyst developed by Venturello and D’aloisio [69] or by blending ESO with soybean oil in different compositions. Moreover, this study contained a discussion on the complex calculus for evaluating the molecular weight between the two cross-linked junctions (Mx). The complexity of the calculus derives from the combined presence of intra and inter-molecular chemical bonds in the thermoset network. The network plateau modulus (GN), *T*_g_, and Mx of the obtained thermosets were calculated by DMA analysis. Those values were analyzed and compared as a function of the different fepoxy formulations. The authors also compared the mechanical properties of copolymerized films made of vegetable oil and divinylbenzene (DVB) with pure EVO films. They also studied the thickness of the EVOs films as a function of the irradiation time, and they evaluated the post-curing effect on the properties of the films [70].

Other than using epoxidized oils alone, other authors have reported the combination of EVOs with petroleum-based epoxy derivatives. For example, in 1994, Salleh et al. reported the successful cationic photo-curing of epoxidized palm oil with cycloaliphatic diepoxide to create thermally stable films [71]. Thames et al. also studied mixed formulations. They combined another type of epoxidized oil: castor oil glycidyl ether (COGE) with a cycloaliphatic epoxide (UVR 6100) using triarylsulfonium hexafluorophosphate salt as PhI. Coatings containing up to 40% of COGE showed better water resistance, gloss, gloss retention, and flexibility than UVR 6100 alone [72].

In 2001, Decker et al. mixed epoxidized soyabean oil (ESO) with either bis-cycloaliphatic diepoxide (BCDE) or diglycidylether derivative of bisphenol A (ADE). Two different soybean oil samples with 2 or 3 epoxy groups in the molecule, namely ESO-2 and ESO-3, were epoxidized with paracetic acid. They were then combined with ADE or BCDE to reduce the viscosity of formulations and to enhance the UV-curing speed. The obtained films showed improved chemical resistance. In this work, an interpenetrating polymer network (IPN) was also developed mixing epoxide with acrylate monomers (hexanediol diacrylate or trimethylolpropane triacrylate) in the presence of both radical and cationic photoinitiators [73].

More recently, modified soybean oil with urethane branched-chain terminated by epoxy group (SBO-URE) was mixed at different weight percentages (10, 15, and 20 wt%) with 3,4-epoxycyclohexylmethyl 3,4-epoxycyclohexanecarboxylate (ECC) and photocured. Three different triarylsulfonium salts were evaluated as photoinitiators. However, low final epoxy group conversion was obtained. This result was ascribed to the short-wavelength light absorption of the PhI in the range of emission of the UV lamp used. Conversions above 60% were achieved by adding small amount of thioxanthone derivatives as photosensitizers. The long flexible aliphatic chain of SBO-URE contributed to increasing the flexibility and enhancing thermal stability of the ECC network [74].

Another example of a mixture of epoxidized oil and cycloaliphatic diepoxide was analyzed by Wan Rosli et al. In this work, epoxidized palm oil and cycloaliphatic diepoxide (Cyracure 6105) were mixed in different concentrations and UV-cured using radical, cationic or hybrid initiator. The addition of two types of vinyl ether monomers into the formulations was also investigated. It was found that increasing EPO content resulted in films with higher hardness and gel content [75].

The photocuring of norbornyl epoxidized linseed oil (NELO) combined with three different divinyl ether reactive diluents were investigated by Chen et al. NELO with an enhanced number of norbornyl epoxide functional groups was obtained via high-temperature and high-pressure Diels–Alder reaction followed by an epoxidation involving H_2_O_2_ and a quaternary ammonium tetrakis(diperoxotungsto) phosphate(3,2) as a catalyst. The effect of type and concentration of divinyl ether were investigated and compared with UV-cured films obtained from ESO and cycloaliphatic epoxide (UVR-6110) [76].

Composites made of a cationic photocured mixture of EVOs and synthetic epoxy resins were also developed in 1997. Crivello et al. reported epoxidized linseed and epoxidized soybean oils blended with 3,4-Epoxycyclohexylmethyl 3′,4′-epoxycyclohexane carboxylate to create fiberglass-reinforced composites. The formulations containing woven fiber-glass were cured with triarylsulfonium hexafluoroantimonate salts both under UV and visible (solar) light [77]. Later, Crivello developed an ELO or ESO visible-light composites using diaryliodonium salt and curcumin as a photosensitizer. Four-layer glass cloth laminates were successfully cured after 10 min of solar irradiation [78].

Formulations containing epoxidized sunflower oil and different amount of modified montmorillonite were cured with a combination of two PhI: Irgacure 250 (cationic) and Darocure 1173 (radical). The radical PhI was added into the formulations to allow the crosslinking of the remaining unsaturations still present in the epoxidized oil chains. The authors focused their attention only on the different UV radiation energy doses needed to obtain the nanocomposites [79].

Another type of composite was obtained from a mixture of ESO and (R)-12-hydroxystrearic acid (HSA). During the crosslinking reaction, HAS nanofibers form clusters able to improve the tensile strength, storage modulus and tensile modulus of the nanocomposite [80].

Epoxidized soybean oil was also employed to create UV-curable solvent-free pressure-sensitive adhesives (PSA). PSA was obtained from the copolymerization of ESO, dihydroxyl soybean oil (DSO), and rosin ester (Sylvalite). The obtained adhesive proved to be thermally stable and showed superior peel strength than commercial tapes [81].

A new type of silyl radical chemistry was developed in 2010 to promote the free-radical cationic polymerization process (FRPCP) of ESO, ELO, and limonene dioxide. In this study diphenyliodonium hexafluorophosphate (Ph_2_I^+^) was mixed with phenylbis(2,4,6-trimethylbenzoyl)phosphine oxide (BAPO), 2,4,6-trimethylbenzoyl-diphenyl phosphine (TPO) or isopropylthioxanthone (ITX). Tris-(trimethylsilyl)silane (TTMSS) was used as additives in all the formulations to avoid oxygen inhibition. The mixtures were irradiated with different wavelengths, including visible light. Tack-free coatings were always obtained [82,83,84].

### 3.2. Epoxidized Cardanols

In recent years, epoxidized cardanol derivatives (ECD) are gaining increasing attention since they possess unique properties deriving from the combination of a hard-aromatic structure with a long flexible alkyl chain (Figure 6).

In 2009, the cationic UV-curing of a commercially available epoxidized cardanol: Lite 2513HP (Cardolite^®^) was firstly reported. In this work, ECD (10 wt%) was added into a cycloaliphatic epoxy resin (UVR 6110) formulation and cured with triarylsulfonium hexafluoroantimonate salt under a Dimax lamp with an intensity of 35 mW/cm^2^. The monofunctional ECD lowered the viscosity of the formulation and increased the mobility of the chains, which resulted in an enhanced monomer conversion. The formulations studied also involved two different hydroxyl-functional reactive diluents: Tone 301 and UVR 6000. Those diluents were added to increase the conversion though chain transfer mechanism, while the presence of ECD prevented the excessive water-hydroxyl interaction able to inhibit the cationic photopolymerization. The intrinsic hydrophobicity of ECD derives from the long aliphatic chain. This work proved the feasibility of utilizing ECD as a “humidity blocking” reactive ingredient for the photo-curing of different formulations [85].

Later, in 2017 Kanehashi described the epoxy prepolymer preparation and UV-curing starting from distilled cashew nut shell liquid (CNSL) and cardanol. The epoxidation reaction was performed with epichlorohydrin and potassium hydroxide. The epoxy CNSL (ECP) was photocured with three different PhIs: bis (4-tert-butylphenyl) iodonium hexafluorophosphate (BIP), tri-p-tolylsulfonium hexafluorophosphate (TSP) and 4-isopropyl-4′-methyldiphenyliodonium tetrakis(pentafluorophenyl)borate (IIPB). The epoxy cardanol alone was not able to create a crosslinked network. The type and concentration of PhIs were evaluated in different CNSL formulations. The fastest CNSL curing time (60 sec) was reached with 4 wt% of BIP, suggesting that the iodonium cation is more suitable to cure this prepolymer. This result could be attributed to the enhanced electron-accepting property of iodonium cation with respect to the sulfonium one. To further boost the reaction, the influence of different photosensitizers on the ECP/BIP system were also studied. Moreover, films made of epoxy cardanol blended with ECP in different weight percentage proportions: 30:70; 40:60; 50:50 were also examined. The thermal stability of the obtained films was higher than the one obtained with the same prepolymer after thermal curing with an amine hardener [86].

In 2019 NC-514S (Cardolite^®^) epoxidized cardanol was combined with micro fibrillated cellulose (MFCs) to obtain fully-biobased composites. Formulations made of NC-514S and different amounts of MFCs were photocured using (4-methylphenyl) [4-(2-methylpropyl) phenyl]-hexafluorophosphate(1-) as a cationic initiator. The photo-reactivity of the formulations containing NC-514S and different amounts of PhI (5, 15, and 22 wt%) was evaluated by FTIR analysis. In the formulation containing 5 wt% of PhI, the epoxy ring peak disappeared after 3 min of irradiation, while 1 min was sufficient to obtain the same results with other formulations. The glass transition temperatures of the crosslinked NC-514S obtained from DSC analysis ranged from −4 to −7 °C. The addition of MCFs resulted in the formation of transparent flexible films [87].

Recently, three types of epoxidized commercially available cardanols: NC-513, LITE 2513HP, and NC-547 were further epoxidized with acetic acid, H_2_O_2_, and Amberlite^®^ IR-120. The obtained products were UV-cured with 2 wt% of triarylsulfonium hexafluoroantimonate salts. Their reactivity was investigated by FTIR analysis. A high epoxy group conversion was reached in all cases (64–72%). The *T*_g_ values (calculated from DSC analysis) of the cross-linked films ranged from 25 to 53 °C. Those values are relatively high if compared with other previously fabricated commercial epoxy cardanol thermosets. The final thermo-mechanical properties of the crosslinked networks were evaluated by DMTA analysis, and the results were compared and discussed according to the structure and epoxy equivalent of the starting monomers [88].

In 2015, Cheon et al. firstly reported the electron-beam-induced cationic polymerization of diepoxidized cardanols (DEC) in the presence of triarylsulfonium hexafluorophosphate or triarylsulfonium hexafluoroantimonate photoinitiators. The epoxy conversion and glass transition temperature of the obtained networks were evaluated, varying the electron beam dosage. Some of the obtained crosslinked networks reached epoxy conversions of 55–60% (calculated from FTIR analysis), while the maximum *T*_g_ was −2.9 °C (calculated from DSC analysis). Those values are comparable to ones obtained with UV curing technology [89,90].

### 3.3. Others

In addition to oils and cash nutshell liquid, there are different biobased materials that have been successfully epoxidized, and subsequently, UV cured. Figure 7 presents other biobased sources of epoxidized derivatives.

Starch is one of the most abundant and readily available carbohydrates in the world. However, its applicability in thermoset applications is limited by its inadequate mechanical properties. Recently different starch modifications have addressed the interest of the scientific community to partially replace petroleum-derived polymers.

In 2003, Han et al. grafted glycidyl methacrylate onto starch (starch-g-GMA) using ceric ammonium nitrate in an aqueous solution. The glycidyl groups were left as a reactive pendant, while the acrylic groups were involved in the free radical grafting mechanism. The starch-g-GMA (~13 wt%) was then copolymerized with a cycloaliphatic diepoxide (CAE). The formulations were UV cured using triphenylsulfonium hexafluorophosphate as a cationic photoinitiator under a fusion lamp with 7 m/min conveyor speed. In order to complete the cross-linking reaction, the obtained coating was subjected to a post-curing process. The obtained copolymer films showed enhanced flexibility with respect to the CAE alone. This result can be attributed to the presence of many hydroxyl groups in the starch chains, which can act chain transfer agents in the cationic reaction [91].

Not only carbohydrates themselves have been used in cationic UV-curing, but also other compounds deriving from polymeric carbohydrates. For example, Cho et al. focused their attention on three different furanic compounds deriving from cellulose and hemicellulose biomass: 2-furan methanol (FM), 2,5-furan dimethanol (FDM) and 2-furan carboxylic acid (FCA). This group described the epoxidation of those monomers and evaluated their applicability in the cationic-photocuring adhesive field. The synthesized monomers were: mono-epoxide (FmE), di- epoxide (FdE), and bisphenol A-like (bFdE) furanic compounds (Figure 5). The reaction kinetics of FmE and FdmE were followed by photo-differential scanning calorimetry and compared with the one obtained by phenyl glycidyl ether (PGE), a petroleum-based resin. In this study, two types of PhIs were used, and their amount was optimized: (4-Methylphenyl) [4-(2-methylpropyl)phenyl] iodonium hexafluorophosphate and triphenylsulfonium hexafluoroantimonate salts. The strength of the bio-based adhesives was evaluated with a tensile-shear test on the bonded joint of two polycarbonate plates. The most promising monomer was the FmE, since the FmE-based adhesive showed higher tensile-shear strength than the PGE [92].

Nameer et al. in 2018 reported the epoxidation and photocuring of 2,5-Furandicarboxylic acid, a biobased furan compound. In this study, the obtained diglycidyl furan-2,5-dicarboxylate (DGFDC) was copolymerized with three different epoxidized fatty acid methyl esters (EMX). Those EMX were obtained from the transesterification of epoxidized linseed oil. Different monomer mixtures were then UV-cured using 2 wt% of p-(octyloxyphenyl)phenyliodonium hexafluoroantimonate. As a comparison, formulations containing DGEBA instead of DGFDC were also studied. Depending on the monomer mixture, the photocuring was performed at 85 °C with an irradiation time ranging from 30 to 60 min. The reaction kinetics was followed with real-time FTIR, and the kinetic parameters were determined from the integration of the reaction enthalpy calculated from DSC analysis. In this investigation, the structure-property relationships of the obtained networks were also analyzed. The addition of the DGFDC monomer increased the network rigidity and mechanical stability, while the addition of EMX monomer enhanced the flexibility and hydrophobicity [93].

Other compounds widely occurring in plants are monoterpenes. Most of them are volatile by-products of processing of plants. In 1995, Crivello and Yang firstly described the epoxidation of three isomeric monoterpenes and their cationic photopolymerization. The starting terpenes were: α-terpinene, γ-terpinene, and limonene. Their epoxy derivatives were UV-cured under a fusion lamp with 0.5 wt% of (4-decyloxypheny1) phenyliodonium hexafluoroantimonate as PhI. Both γ-terpinene diepoxide and limonene diepoxide were demonstrated to be very reactive towards cationic photopolymerization. Instead, a thermoset could not be generated by UV-curing of isoascaridole [94].

Later, Park et al. deepened the study concerning the photocuring of monoterpenes focusing on limonene 1,2-oxide (LMO) and α-pinene oxide (α-PO). These compounds were photocured using (4-n-undecyloxyphenyl)phenyliodonium hexafluoroantimonate or S(4-ndecyloxyphenyl)-S,S-diphenylsulfonium hexafluoroantimonate as PhIs. However, homopolymerization studies highlighted the presence of many side reactions leading to the formation of nonpolymeric products. For this reason, these two biobased monomers were studied as comonomers with different commercially available epoxy petroleum-based resins. Moreover, the polymerization of epoxidized linseed oil in the presence of LMO and α-PO was also evaluated. In general, the addition of these monomers lowered the viscosity of the formulation, reduced the induction period, and enhanced the polymerization rate leading to thermosets with improved mechanical properties [95].

In 2019, Breloy et al. reported the successful cationic photocuring of di-epoxy limonene (Dipentene dioxide-DPDO) under visible light. This result was achieved using β-carotene as natural photosensitizer of iodonium salt. In this work, limonene 1,2 epoxide (Lim) bearing in its structure both an epoxy ring and an allyl group was also studied. The scope of this investigation was to simultaneously involve Lim in both cationic photopolymerization and thiol-ene click reaction. Therefore, formulations containing different amounts of trithiol (TT), Lim, β-carotene, and iodonium salts were analyzed. However, those systems failed to produce cross-linked networks. The addition of DPDO into the formulations was necessary to obtain tack-free coatings. Furthermore, to give the network antibacterial properties, eugenol was incorporated into the systems, adjusting the TT amount accordingly [96].

In 2020, a terpenoid-like nopol produced from β-pinene, was selected as the starting material for producing three different cycloaliphatic epoxy monomers (Figure 8a); one bifunctional (N1) and two trifunctional (N2 and N3). Those products were subsequently photocured with decyloxyphenyl phenyl iodonium hexafluoroantimonate. As previously mentioned in Section 2.2, the cycloaliphatic epoxy monomers undergo a swift cationic polymerization reaction since their epoxy rings are highly strained. However, in this case, the final epoxy group conversion reached only 62%, 58%, and 54% for N1, N2, and N3, respectively. These results could be attributed to early vitrification of the networks. The overall properties of the trifunctional epoxy thermosets were higher than the bifunctional one, in terms of both glass transition temperature and thermal stability [97].

As previously mentioned in Section 3.1, Noè et al. also reported the synthesis and cationic UV-curing of diglycidylether of vanillyl alcohol (DGEVA) (Figure 8b) and phloroglucinol tris-epoxy (PHTE) (Figure 8c). Those monomers were photocured using triarylsulfonium hexafluoroantimonate salts as PhI under a Fusion lamp. In all cases, free-standing films were obtained. The reaction conversions were studied by FTIR analysis.

The epoxy group conversion of DGEBA (60%) was lower than the one obtained with PHTE (80%) even though the first aromatic monomer is difunctional and the second one is trifunctional. This can be explained by considering the higher hydroxyl group content in PHTE compared with DGEBA; thus, they generate a more flexible ether structure via AM mechanism, delaying the vitrification and leading to higher epoxy group conversion. This mechanism can also explain the increased *T*_g_ of DGEBA around 82 °C, with respect to the PHTE of about 75 °C [67].

In 2020, Nguyen et al. reported the photocuring of diglycidyl resorcinol ether (RDGE). Resorcinol can be extracted from lignocellulosic biomass and constitutes another source of aromatic biobased compounds. RDGE formulations were cured with (bis[dodecylphenyl] iodonium hexafluoro-antimonate in a UV-oven equipped with six low-intensity (1 mW/cm^2^) lamps. Benzophenone was used as a photosensitizer. With the interest of obtaining photoinduced bulk resin samples, the irradiation time was varied from 2 to 2.5 h. However, the obtained FTIR conversions were below 60%. In order to overcome this problem, different post-curing treatments were investigated. A full epoxy group conversion and a very high *T*_g_ (99 °C) were obtained after 2 h of irradiation and 3 h of post-curing at 150 °C. Additionally noteworthy are the flexural mechanical properties obtained by this optimized resorcinol system (A1). For example, the A1 flexural modulus value (4.5 GPa) is higher than the values obtained for different petroleum-based resins tested for comparison. For the first time, a fully biobased epoxy thermoset with remarkable mechanical properties able to compete with the petroleum-derived resins was presented. Nevertheless, even if the monomers were photocured, the long irradiation time required represents a drawback that can limit the applicability of the obtained materials [98].

## 4. Conclusions

Fossil-fuel resources have aroused great concern for both environmental and raw material scarcity issues. Here, bio-based epoxy resins have great potential in replacing petroleum-based resin for fabrication of thermosets. Among the curing processes, cationic UV curing has numerous advantages over conventional thermal curing. In particular, the photopolymerization has low volatile emissions, fast curing speed, and no shrinkage. Moreover, since this technique does not require any additional heating, it can be cost-effective. However, as it has been clearly presented in this review, the properties achieved by biobased UV-cured thermosets so far can only match the requirements needed in the fields of coatings, films, or adhesives. The use of biobased epoxidized monomers in the cationic UV-curing process remains a challenging research area with promising prospects.

## Figures and Tables

**Figure 1 polymers-13-00089-f001:**
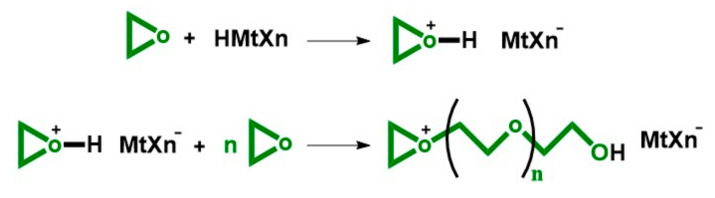
Mechanism of the cationic ring opening polymerization of epoxide.

**Figure 2 polymers-13-00089-f002:**
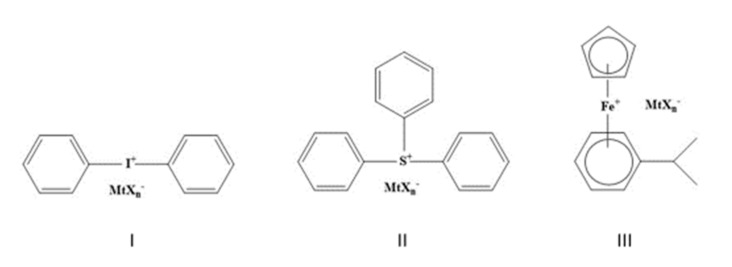
Structures of most common cationic photoinitiators.

**Figure 3 polymers-13-00089-f003:**
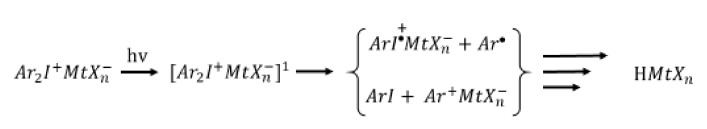
Simplified photodecomposition of a general diaryliodonium salt.

**Figure 4 polymers-13-00089-f004:**
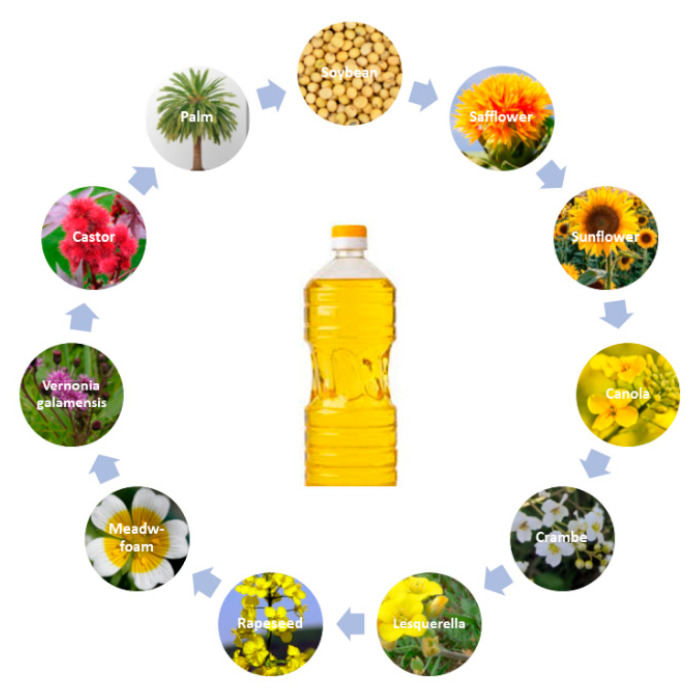
Common botanic oils used in cationic UV curing systems.

**Figure 5 polymers-13-00089-f005:**
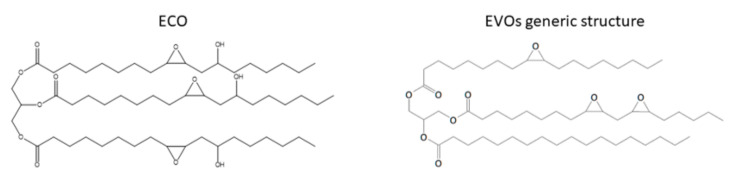
ECO structure and generic EVOs structure.

**Figure 6 polymers-13-00089-f006:**
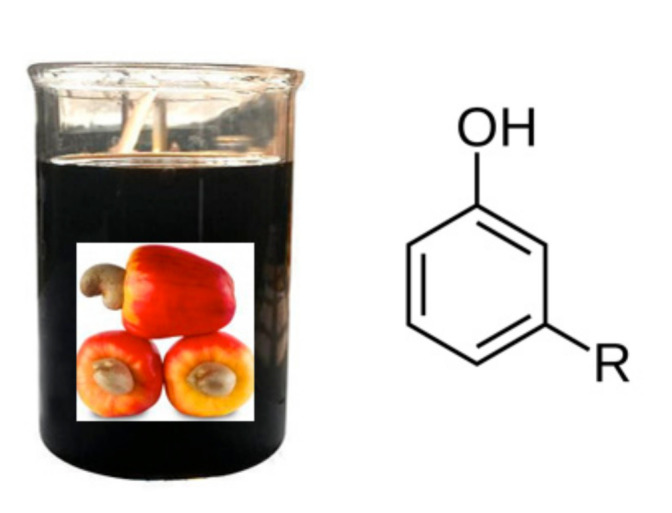
Cash nut shell liquid and a general cardanol structure, where R is an alkyl chain.

**Figure 7 polymers-13-00089-f007:**
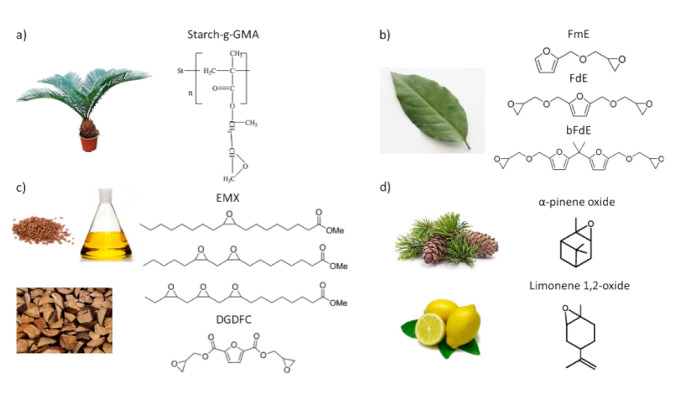
Some examples of epoxidized monomer/polymer structures derived from different biobased resources. (**a**) Starch-g-GMA; (**b**) epoxidized furanic compounds FmE, FdE, and bFdE; (**c**) diglycidyl furan-2,5-dicarboxylate (DGFDC) and different epoxidized fatty acid methyl esters (EMX); (**d**) α-pinene oxide and 1,2-oxide (LMO).

**Figure 8 polymers-13-00089-f008:**
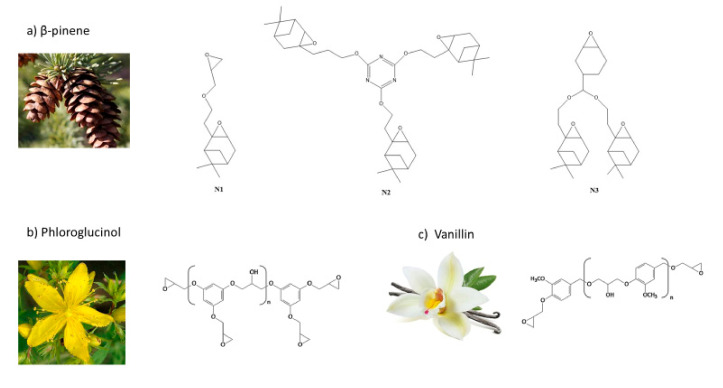
Epoxidized monomers derived by (**a**) β-pinene, (**b**) phloroglucinol and (**c**) vanillin.

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
