# Peer review of "Cationic UV-Curing of Epoxidized Biobased Resins"

_polymers, 2020, doi:10.3390/polym13010089_

Round 1
Reviewer 1 Report
The review is very well written and logically structured. However, there is one common drawback - the effects achieved, as well as the properties obtained as a result of the corresponding modifications, are mainly described in general terms, without data, tables, diagrams, etc.
Author Response
We really appreciate the very positive feed-back of the reviewer who reported that our review "is very well written and logically structured". It is difficult to summarize the overall literature data reported in the review in a form of a Table, as the reviewer asked. In fact, per each group of bioderived epoxides the different authors reported different characterization of the crosslinked materials and it is complex to summarize what is clearly reported in the text with a simple Table. In the text a comparison of epoxy group conversion and Tg values achieved are described. We do not think that a Table would make the review more readble.
Reviewer 2 Report
The work developed by Noé et al. is very interesting and complete. However, most of the work sited is old. Indeed, very little references are from the last 5 years. Since this subject is not exatly new it would be important that more information would arise from recent work. Please introduce studies made in 2019 and 2020 in your data.
Generally the research is very interesting, and the ifnormation is both to the point and scientidically sound. The graphical representations are also very helpful. Once more current research is introduced in the manuscript is can be published.
Author Response
We gratefully acknowledge the reviewer for the very positive comments. We have further controlled the literature related to the year 2019 and 2020. We have therefore added, in the revised version of the paper 3 new reference from 2019 (1 new paper) and 2020 (2 new paper). The topic is very interesting but also not extensively convered in literature yet. We think this revised version of the review can be considered now complete.
Reviewer 3 Report
The review "Cationic UV-curing of epoxidized biobased resins" is well written, easy comprehensible and covers all the aimed issues.
It has to be checked for minor editing errors as:
line 54. [15,16] instead of [15], [16].
line 84. Figure 1 instead of Scheme 1. Figure 3 instead of Scheme 2 etc.
Besides these minor issues, I have no comments and suggestions.
I congratulate the authors for a quality submission.
Author Response
We really thanks the reviewer very much for the very good comments. We have correct the few mistakes present in the text.
Round 2
Reviewer 2 Report
The authors followed the reviewers' recommendations and added more recent research. Even though little the effort was good. The manuscript is now ready for publication.